# Role of Tumor-Derived Extracellular Vesicles in Glioblastoma

**DOI:** 10.3390/cells10030512

**Published:** 2021-02-28

**Authors:** Yunping Chen, Yan Jin, Nan Wu

**Affiliations:** 1Laboratory of Medical Genetics, Harbin Medical University, Harbin 150081, China; chenyunping@hmudq.edu.cn; 2Department of Pharmacology, Harbin Medical University-Daqing, Daqing 163319, China; 3Key Laboratory of Medical Genetics (Harbin Medical University), Heilongjiang Higher Education Institutions, Harbin 150081, China

**Keywords:** glioblastoma, extracellular vesicles, tumor biomarker

## Abstract

Glioblastoma (GBM) is the most common primary central nervous system tumor and one of the most lethal cancers worldwide, with morbidity of 5.26 per 100,000 population per year. These tumors are often associated with poor prognosis and terrible quality of life. Extracellular vesicles (EVs) are membrane-bound nanoparticles secreted by cells and contain lipid, protein, DNA, mRNA, miRNA and other bioactive substances. EVs perform biological functions by binding or horizontal transfer of bioactive substances to target cell receptors. In recent years, EVs have been considered as possible targets for GBM therapy. A great many types of research demonstrated that EVs played a vital role in the GBM microenvironment, development, progression, angiogenesis, invasion, and even the diagnosis of GBM. Nevertheless, the exact molecular mechanisms and roles of EVs in these processes are unclear. It can provide the basis for GBM treatment in the future that clarifying the regulatory mechanism and related signal pathways of EVs derived from GBM and their clinical value in GBM diagnosis and treatment. In this paper, the research progress and clinical application prospects of GBM-derived EVs are reviewed and discussed.

## 1. Introduction

Gliomas are primary tumors of the brain produced by glial stem cells or progenitors [1]. Gliomas are generally classified into four grades adopts the World Health Organization (WHO) classification system, with grade I and II being benign and III and IV being malignant. Grade IV glioma is also known as glioblastoma (GBM). GBM is the most deadly glioma type [2], accounting for 70–75% of all diffuse gliomas, with a median overall survival of only 14–17 months [3].

Many studies indicated that few available therapies could significantly improve survival chances of patients with GBM [4,5,6]. The histopathological definition of GBM is based essentially on the presence of tumor cells considered to be of neuroglial origin and on the examination of neovascularization and necrosis. At the molecular level, GBM is typically characterized by a lack of isocitrate dehydrogenase 1 or 2 (IDH1 or IDH2) mutations and mutations in genes regulating RTK/RAS/PI3K, p53, and RB. Amplifications and mutations of the epidermal growth factor receptor gene (EGFR) are found in 40% of the GBM [1,7,8,9]. GBMs are usually also stratified by the DNA repair enzyme O6-methylguanine-DNA methyltransferase (MGMT) promoter methylation, as well as TERT (telomerase reverse transcriptase) promoter mutations, mutations/deletions in PTEN (phosphatase and tensin homolog) and nowadays, many efforts are made using whole-genome methylation data [10,11,12].

GBM’s fast deterioration, drug resistance and high recurrence rate attribute to several kinds of elements, including its rapid proliferation, extensive invasion, and genetic heterogeneity within the tumor. Moreover, the poor prognosis of GBM patients also partly results from a lack of understanding of molecular mechanisms and timely diagnosis and sensitive therapeutic monitoring tools [13]. Conventional therapies, including surgery, radiotherapy and typically chemotherapy with temozolomide, have not been conducive to major improvements in the survival outcomes of patients with GBM [14,15,16].

In the past several years, an increasing number of studies have indicated that extracellular vesicles (EVs) and the contents of EVs played an important part in the initiation, progression and diagnosis of GBM [17,18,19,20]. At present, attention has been focused on how EVs playing a role in mediating cellular communication in the tumor microenvironment of various cancers, including GBM. The field of EVs has attracted increasing attention in recent years. Many reviews have been published, which have contributed to the development of this field and attracted more researchers to devote themselves to this field [21,22]. In order for more researchers and those who are new to this field to have a quick and concise understanding, we have carried out this review, which mainly includes the formation, release, uptake, structure, function, isolation and purification of EVs; GBM-derived EVs play a variety of roles in GBM tumorigenesis, microenvironment, angiogenesis, immune response, metastasis, invasion, subtype and chemotherapy resistance.

## 2. Formation, Release and Uptake of EVs

EVs were first discovered in 1983 in mature sheep reticulocytes. They were originally named “exosomes” and thought to be cell debris in 1989 [23]. More than a decade later, it was reported that B lymphocytes and dendritic cells could secrete EVs, indicating that EVs had potential immunomodulatory effects and were regarded as carriers of antitumor immunity [24,25]. The production of EVs involves the double entrainment of the plasma membrane and the formation of intracellular MVBs [26].

Early EVs are originally formed by phagocytosis of cell membranes [27]. They gather molecular substances, such as target proteins, RNA and DNA along the motion path, furthermore, processing, embellishing and classifying these substances. The formation of EVs is primarily divided into endosomal sorting complex required for transport-dependent (ESCRT)-dependent and ESCRT-independent pathways [28,29]. Beyond that, carrying out numerous biologic functions, especially in intercellular communication, accumulated evidence manifested that a few biological substances in EVs like proteins, microRNAs and lncRNAs were bound up with the nosogenesis of many malignant tumors. Furthermore, EVs could be used as promising biomarkers for tumor diagnosis, treatment and the prognosis of survival [30,31,32].

It has been confirmed that all cell types secrete EVs, which play a vital role in cell–cell communication under physiological and pathological conditions [33]. EVs participate in the interchange of genetic information and materials in cell-to-cell communication [34]. They play an important part in sustaining many of the significant physiological processes, such as cell growth, development, differentiation, and apoptosis [35]. More and more research indicated that abnormal secretion and dysfunction of EVs had a significant effect on the genesis, development and therapy of human malignant tumors [36,37,38].

## 3. Structure and Function of EVs

EVs is a collective term that covers the various subtypes of membrane structures released by cells. EVs include species of exosomes with a diameter of 30–150 nm, microvesicles with a diameter of 100–1000 nm and apoptotic bodies with a diameter of 800–5000 nm [39]. EVs carry lipids, proteins, RNAs (including microRNAs, mRNAs and long non-coding RNAs) and DNAs, which transfer these contents from donor cells to recipient cells, causing changes in the tumor microenvironment (TME) [40].

There is growing evidence that EVs play a critical role in cancer development, including tumorigenesis and metastasis [41]. The continuous expansion, invasion and metastasis of tumor cells rest with the communication between cells in the microenvironment. This communication consists primarily of soluble factors secreted by the cancer cell and stromal cells in the tumor microenvironment, and these stromal cells export particles containing regulatory molecules, which facilitate intercellular communication [42,43]. EVs could be precipitated when the centrifugal force was 100,000 g, or the sucrose density gradient was 1.13–1.19 g/mL [44,45]. EVs are spherical in shape, “cup” or “dish” in the transmission electron microscope and have a clear lipid bilayer structure under electron microscope [25,46]. EVs are released not directly from cell membranes by exocytosis but from multivesicular bodies (MVBs) [47,48]. In the process, EVs carry all kinds of bioactive cargo, including protein, lipids, enzymes, DNA and RNA (mRNA, microRNA and lncRNA) [49]. These EVs are involved in homeostasis, intercellular communication, the mediation of the immune system and inflammation and the delivery of genetic information [50,51]. Furthermore, EVs influence the pathological state of many diseases, such as the occurrence, progression, metastasis of cancer, neurodegenerative diseases, infectious diseases and autoimmune diseases [37,49] (Figure 1). EVs play a vital role in the development, progression and drug resistance of various tumors, including GBM [52,53]. Moreover, EVs contribute to maintaining the differentiation of glioma stem cells and influencing angiogenesis, TME and the immune defense system of GBM [54,55]. Studies have shown that EVs increase cell invasiveness, proliferation, migration potential, invasiveness and therapeutic resistance [56,57,58].

## 4. Isolation and Purification of EVs

To explore the function and clinical application of EVs, the first and foremost is to isolate EVs from a large number of cells. The techniques used for EVs separation should be efficient and able to separate EVs from a wide variety of samples [59,60,61]. To check the characteristics of separated EVs, the size, morphological characteristics, distribution, number and components of EVs are measured using optical and non-optical technologies [62,63,64]. Several techniques put forward the enrichment of EVs, such as differential centrifugation, density-gradient ultracentrifugation, size-exclusion chromatography, gel filtration, flow field fractionation, commercial kits using polymer-based precipitation and immunoaffinity-based capture to achieve the purification of tumor-derived exosomes [65,66].

## 5. Glioblastoma-Derived EVs

A great amount of evidence indicates that EVs have significant effects on the formation and metastasis of GBM. Moreover, the substances in EVs are also used for clinical diagnosis and treatment. GBM-derived EVs are severely involved in tumorigenesis, microenvironment, angiogenesis, immune response, invasion, subtype and chemoresistance by transferring oncogenic proteins and nucleic acids.

### 5.1. Tumorigenesis

The important characteristic of GBM is its strong infiltration ability, which leads to the formation of satellite tumors in healthy brain parenchyma. However, it is not able to be surgically removed, which is one of the factors for the early recurrence of GBM patients [67]. Tumor-derived EVs are conducive to the proliferation, migration and invasion of GBM cells [68]. EVs are important carriers of oncogenic factors involved in the development of GBM [19,69].

A recent study has suggested that overexpression of miR-34a in EVs derived from human bone marrow mesenchymal stem cells (hBMSC) inhibited the proliferation and invasion of GBM cells [70]. The EVs secreted by GBM-associated macrophages (GAMs) contained a large amount of miR-21, which was a well-known cancer-promoting microRNA. Moreover, pacritinib, an oral tyrosine kinase inhibitor, could inhibit the occurrence of GBM by regulating signal transducers and activators of transcription 3 (STAT3))/miR-21/programmed cell death 4 (PDCD4) signaling [71].

One research showed that glioma-derived EVs affected M2 macrophage polarization under hypoxia, thus promoting the immunosuppressive microenvironment formation. In addition, researchers found that miR-1246 was abundant in cerebrospinal fluid (CSF) of GBM patients, which may serve as a new biomarker for diagnosis, treatment with targeted miR-1246 may be helpful for therapy of GBM [72]. MiRNA transport regulated by EVs from CD133^+^ U87 cells performed a vital part in adjusting pro-angiogenic responses and cell proliferation [73].

The serum EVs miR-301a level in glioma expressed its cancer biology and pathological variations; what is more, it could be a new biomarker and prognostic indicator for glioma diagnosis [17,74]. One recent study suggested that miR-301a in exosomes secreted by hypoxic glioma cells activated the Wnt/β-catenin signal pathway by targeting TCEAL7 and improved radiotherapy resistance and that TCEAL7 was a confirmed GBM suppressor gene [75].

EVs isolated from the serum of patients with GBM were injected with normal epithelial cells, causing gliomas in mice. When EVs isolated from the serum of patients with GBM were injected simultaneously with normal epithelial cells, the mice developed gliomas, indicating an underlying mechanism for EVs in GBM tumorigenesis [76]. We summarized EVs’ microRNAs that played a role in GBM tumorigenesis (Table 1).

### 5.2. Role of EVs in GBM Microenvironment

It is well-known that the tumor microenvironment (TME) plays a pivotal role in all aspects of GBM development. A recent study suggested that exosomes participated in the disease course of GBM and even played an important role in the reconstruction of TME [77,78]. EVs acted as a key mediator for tumor development and maintenance by regulating TME [79]. EVs mediated the communication between GBM cells and stromal cells, including monocytes, macrophages, mast cells, microglia, T cells, astrocytes and oligodendrocytes [80]. Hypoxia and GBM influenced the polarization of M2 macrophages through EVs, thereby promoting the formation of an immunosuppressive microenvironment [72].GBM tumor cell-derived EVs modulated different cellular and extracellular components of the tumor microenvironment to promote the development and progression of GBM [81]. MiR-301a in EVs secreted by hypoxic GBM cells could be transferred to normal oxygen culture cells to change the cell microenvironment, eventually leading to increased radiation resistance [75]. EVs secreted by GBM gave rise to an immunosuppressive response. STAT3 was a component of GDE and mediated the immunosuppressive switch. The data showed that glioblastoma stem cell-derived exosomes (GDEs) were factors released by GBM and effective regulators of the GBM immunosuppressive microenvironment [82]. Therefore, EVs secreted by GBM played an important role in the microenvironment regulation of GBM.

As an information carrier, glioblastoma stem cells (GSC) EVs mediated the dedifferentiation of non-GSC glioma cells into GSCs by activating Notch1 signal to transmit Notch1 protein, thus enhancing the stemness and tumorigenesis of non-GSC glioma cells [83]. The research described that Semaphorin 7A (SEMA7A) was exposed to the surface of patient-derived glioma-associated stem cells (GASC) EVs and increased the activity of GSC through the interaction between integrin β1 and GSC in the microenvironment [84].

GBM cells derived cancer-causing lncRNA-SBF2-AS1 EVs to reconstruct tumor microenvironment and accelerate tumor chemotherapeutic resistance [85]. EVs secreted by glioma cells activated glycolysis of hBMSCs leading to the transformation of tumor phenotypes. This indicated that disturbing the mutual effect between exosomes and hBMSCs in the tumor microenvironment may be a therapy for glioma [86]. A study elucidated a potential mechanism for glioma recurrence in which normal glioma-associated astrocytes protected MGMT negative glioma cells from temozolomide (TMZ)-induced apoptosis by transferring EVs MGMT mRNA [87].

### 5.3. Role of EVs in GBM Angiogenesis

Angiogenesis, the generation of new blood vessels from existing ones, is essential for the growth and maintenance of tumors [88]. The proliferation, migration, differentiation of endothelial cells and the generation of new blood vessels require the pro-angiogenic factors, anti-angiogenic factors and microRNAs in the EVs secreted by GBM (Table 2). It has been suggested that the targeting of vascular endothelial growth factor (VEGF) from EVs to brain endothelial cells might affect their function to make new blood vessels. Therefore, GBM-derived EVs cargo may be an important part of tumor-induced angiogenesis [89].

### 5.4. EVs and Immune Response of GBM

EVs released by GBM were taken up by microglial cells, which were associated with increased miRNA level, decreased target mRNA and coding protein, leading to increased proliferation of GBM cells and enhanced immunosuppression [94]. GBM regulated the immune system to affect monocytes, macrophages and microglia, leading to GBM’s invasiveness [76]. Interactions between programmed cell death 1 ligand 1 (PDL1) and its receptor programmed cell death 1 (PD1) inhibited the T-cell response. Studies have shown that EVs expressing PDL1 could inhibit antitumor immune responses [95]. GBM EVs blocked T cell activation and proliferation response to T cell receptor stimulation. PD-L1 on GBM-derived EVs might inhibit antitumor immunity [96]. GSC-derived EVs penetrated the cytoplasm of monocytes and induced the recombination of the actin cytoskeleton, which inclined monocytes to the immunosuppressive M2 phenotype and increased the expression of PD-L1. This suggested that EVs were factors released by GBM and effective regulators of GBM-related immune response [82,95]. GBM-derived EVs could modify the phenotype of monocytic cell lineage, including monocytes, macrophages and microglia. EVs altered them to resemble the tumor-supporting phenotypes described by patients [97]. Dendritic cell (DC) vaccine immunotherapy for GBM has displayed significant benefits in animal and early clinical trials. EVs delivered DC vaccine to bring about activation and proliferation of tumor-specific cytotoxic T lymphocytes, destroy immune tolerance and improve immunosuppressive environment [98]. GBM cell-derived EVs LGALS9 in the cerebrospinal fluid played a major regulatory role in the progression of GBM by inhibiting DC antigen presentation and cytotoxic T cell activation [99]. Moreover, antigen-presenting cells could reduce the ability of the immune system, thereby preventing the production of specific immune responses [100]. The serum EVs and cytokines in peripheral blood of GBM patients played a systemic role through immune regulation, which exceeded the limits of the central nervous system [101].

### 5.5. EVs and GBM Invasion

Substantial invasive capacity is one of the key characteristics of GBM [102]. EVs can reshape the extracellular matrix and promote GBM invasion. It has been reported that GBM-delivered EVs promoted GBM invasion offering the necessary stimulation for radiating GBM cells [103,104]. GSCs-derived EVs significantly increased proliferation, neurosphere formation, invasiveness and tumorigenicity of non-GSC glioma cells [83].

Molecules involved in cell migration, including neurotrophic tyrosine kinase receptor type 1 (TrkA), paxillin, focal adhesion kinase (FAK) and the oncogene tyrosine-protein kinase Src. GBM-derived EVs contribute to the activation of these factors [105]. Hypoxic-derived EVs bring about changes in GBM cell biosynthesis and ion regulation channels. Moreover, EVs secreted by hypoxic GBM cells can also promote intercellular communication, induce significant changes in gene expression in adjacent normal oxygen tumor cells, many of which take part in the process of cancer invasion and therapeutic resistance [106]. The novel data suggest that EVs deliver immunoglobulin superfamily protein L1CAM (L1, CD171) to promote GBM cell proliferation and invasion, thereby influencing GBM cell behavior [107]. In conclusion, the above researches suggest that EVs-mediated cell–cell communication may be involved in important mechanisms of GBM invasion.

### 5.6. The role of EVs in GBM Subtype

GBM subtype-specific gene markers are vital to the research of tumor heterogeneity [108]. In high-grade glioma (HGG), different levels of cell populations are derived from different glioma stem cell-like (GSC) subpopulations. It was found that EVs protein transport between different tumor cell subsets provided a means of dynamic transformation and was correlated with HGG heterogeneity [57]. MiR-128-dependent transcriptome on EVs could predict subtypes and prognosis in patients with GBM [109].

EVs cargo is different between GBM subtypes, including neurological neoplasms, neoplasms, classical neoplasms and mesenchymal neoplasms [110]. EVs are carriers of molecular and oncogenic signals present in tumor cell subsets and tumor-associated stroma, as well as mediators of intercellular communication. EVs may also reflect and influence balance at the stem cell level, including oncogenic drivers and regulatory microenvironments [111]. The role of EVs as biomarkers and mediators for GBM may depend on the molecular subtypes and functional status of donor tumor cells, including GSCs [81].

### 5.7. GBM EVs Induce Chemoresistance

An important reason for the poor GBM survival rate is chemoresistance. GBM-derived EVs play an important role in chemotherapy resistance. In GBM, EVs secreted by ptPRZ1-MET fused cells lead to a tumorigenic phenotype and temozolomide resistance [69]. Polymerase I and transcriptional release factor (PTRF) known as Cavin1 in serum EVs contributed to the detection of gliomas were hopeful biomarkers and may be a treatment target point for GBM [112].

Furthermore, it was found that lncRNA SBF2-AS1 in EVs was increased in TMZ-resistant GBM cells, while overexpression of SBF2-AS1 could promote TMZ drug resistance. On the contrary, inhibition of SBF2-AS1 made TMZ-resistant GBM cells sensitive to TMZ [113]. The TMZ-resistant GBM cells were related to miRNAs and EVs [16]. MiR-1238 was incorporated into EVs by TMZ-resistant GBM cells, resulting in a high-level of miR-1238 in EVs, which was absorbed by TMZ sensitive cells and spread TMZ resistance [114].

GBM cells transported miR-93 and miR-193 through EVs to target cyclin D1 to achieve TMZ-induced drug resistance [115]. TMZ-resistant GBM cell-derived EVs delivered miR-151a in an independent manner, so that receptor TMZ sensitive cells developed TMZ chemotherapeutic resistance [116]. Mesenchymal stem cell-derived EVs transport synthetic anti-miR-9 and reverse chemotherapeutic resistance of GBM cells [117].

## 6. Conclusions and Future Perspectives

As a new intercellular communication mechanism, EVs enable GBM cells to acquire different characteristics of tumoral recurrence and progression [118,119]. GBM-derived EVs contain various components, including DNAs, miRNAs, lncRNAs, mRNAs, enzymes, ligands and receptors [20,120]. EVs play an irreplaceable role in maintaining the stability of the human body [121]. Compared with other tumor markers in tissues or body fluids, EVs have higher stability and content, so they have more advantages and significant potential for clinical application in the early diagnosis and therapy of GBM [122]. The ability of EVs to cross the blood–brain barrier (BBB) is an important factor in considering their success in delivering drugs to TME. GBM-derived specific EVs can cross the BBB and circulate in body fluids, so they can be used as non-invasive biomarkers for early diagnosis of GBM [123,124]. GBM-derived EVs could enhance BBB permeability [125].

Research interests in the EVs area are focused on the use of EVs-based non-invasive biomarkers for cancer diagnosis and surveillance and the use of EVs to deliver targeted drugs for GBM treatment. EVs secreted by GBM tumor cells can be detected in the blood, cerebrospinal fluid, and urine of cancer patients [126]. Therefore, EVs can be used as diagnostic and therapeutic monitoring tools for GBM. EV isolated from glioblastoma cell lines contained tumor-specific mRNAs and miRNAs [127,128]. We represent the roles of EVs in GBM in a simplified diagram (Figure 2).

In addition, we highlighted the GBM-derived EVs influencing tumor microenvironment, angiogenesis, microRNAs, tumorigenesis, immune response, subtype and chemoresistance. Moreover, EVs regulate the migration and invasion of GBM cells, which is expected to break new ground for GBM treatment. EVs are similar in size and function to synthetic nanoparticles and have many advantages that make them the most promising targeted drug or gene vectors [129]. EVs may be a more targeted material source for the discovery of biomarkers. Systematic EVs accumulate in the liver, kidney, and spleen. Several types of EVs secreted by cells exhibit target selection, including DCs, B-cells and macrophages [130,131]. Target EVs can be obtained by presenting target molecules, such as peptides or fragments of antibodies that recognize target antigens on the outer surface of EVs. However, the tendency of EVs secreted by most cells to specific cell types is limited, so future research requires targeting strategies for systemically delivered EVs. Moreover, future studies on EVs urgently need to understand the potential of EV-mediated targeted communication to enable EVs as a novel cancer therapy. There is an urgent need to improve the identification, isolation and purification techniques of EVs, and the molecular mechanism of action on diseases is still not very clear. Synthetic EV technology is not mature enough at present.

The field of GBM-derived EVs has attracted more attention in recent years. We hope that more researchers will pay attention to this field on the basis of GBM-derived EVs in future studies so as to jointly solve the important issue of the occurrence and development of GBM. We hope that a growing number of researchers will join us in the research area of EVs, which shows the importance of this area. In conclusion, there is still a long way to go to achieve clinical application before the above problems have been explored and verified, especially the research on GBM diagnosis and treatment is still in the experimental stage.

## Figures and Tables

**Figure 1 cells-10-00512-f001:**
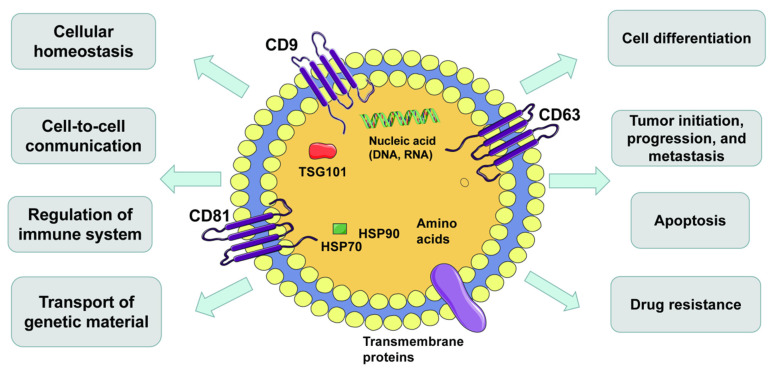
Hallmarks of extracellular vesicles (EVs). EVs are secreted by all kinds of cells and carry DNAs, RNAs, proteins, lipids and metabolites. EVs are important mediators of intercellular communication and affect all aspects of cell biology.

**Figure 2 cells-10-00512-f002:**
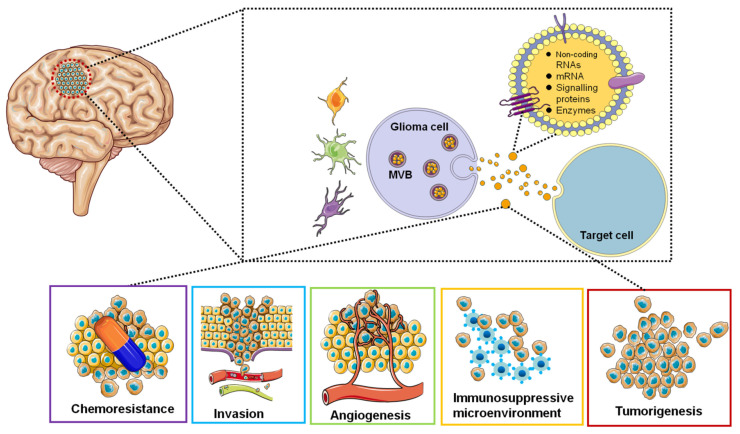
Functions of glioblastoma-derived EVs. EVs are involved in tumorigenesis, microenvironment, angiogenesis, immune response, invasion and chemoresistance by transferring oncogenic proteins and nucleic acids.

**Table 1 cells-10-00512-t001:** Tumorigenesis in EVs’ microRNAs with a role in glioblastoma (GBM).

Molecule	Source	Effect on GBM	Experimental Model	References
miR-34a	hBMSC	Inhibited GBM cell proliferation, invasion, migration and tumorigenesis	GBM cell lines (T-98G, LN229 and A-172)	[70]
miR-21	GAMs	Increased ability to polarize GAMs towards the M2 phenotype	GAMs, U87MG and LN18	[71]
miR-1246	CSF	Induced M2 macrophage polarization		[72]
miR-301a	serum	Promoted the proliferation and invasion of glioma-derived H4 cells	Serum and glioma-derived H4 cells	[17,74]
miR-301a	hypoxic glioma cells	Promoted radiation resistance	U87MG, LN229 and U251	[75]

hBMSC—human bone marrow-derived mesenchymal stem cells, GAMs—GBM-associated macrophages, CSF—cerebrospinal fluid.

**Table 2 cells-10-00512-t002:** Angiogenic microRNAs in EVs with a role in GBM.

Molecule	Source	Effect on CBM	Experimental Model	References
miR-5096	Glioblastoma cells	Increased proliferation and invasiveness	Human microvascular endothelial cells, U87 and U251 cells	[90]
miR-221	Glioblastoma tumor cells	Increased proliferation and migration; increased tumor growth	U87, SHG-44, HEB, U251 cells, nude Mice	[91]
miR-21	glioma stem cells	Increased angiogenesis	Human brain endothelial cells, glioma stem cells	[92]
miR-15b, miR-16, miR-19b, miR-20, miR-21, miR-26a, miR-27a, miR-92, miR-93, miR-320	glioblastoma cells	Increased proliferation and angiogenesis	Human brain microvascular endothelial cells	[93]

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
