# Peer review of "Role of Tumor-Derived Extracellular Vesicles in Glioblastoma"

_cells, 2021, doi:10.3390/cells10030512_

Round 1
Reviewer 1 Report
Response to cells
Manuscript Number: Cells-1072133
Title: Role of tumor-derived exosomes in glioblastoma
The review by Chen et al. reviews the role of tumor-derived EVs (not exosomes!!!, needs to be corrected throughout the text -> see MISEV guidelines) in glioblastoma. The manuscript is well written, yet it needs a thorough work over, as multiple points in the field of EV research are not at the current standard (especially in the nomenclature).
Point to discuss:
- 1 GBMs are usually also stratified by MGMT promotor methylation, as well as TERT mutations, and nowadays a lot of efforts are made using whole-genome methylation data.
- 2. please review the word Exosomes. As seen above.
- 2. EVs should not be characterized simply by their size (MISEV guidelines)
- 2 EVs have been shown to increase proliferation in GBM please discuss Kucharzewska et al, PNAS, 2013 and Ricklefs et al, Cancer Research, 2016
- 3. figure. In the legend DNA should also be menationed, as it is already seen in the figure itself
- 3. the authors should cite the original work and not only the Review [lit. 42/43]
- The authors state variety of cells excrete EVs (not secrete as this is done for cytokines, etc.). But the actual question remains, which cells do not excrete EVs. To my knowledge no cell type has been found to not excrete EVs.
- 4. Metastasis of GBM? This is so unbelievable rare, that I am certain, that there is no data on how EVs might help this process. Please revise.
- 4. GBM sufferer (???) I believe you mean patient.
- 5. Please also discuss the role of GBM EVs in the role of GBM subtype switch (see Godlewski, Stem cell Reports, 2017; Rooj, Cell reports, 2017, Ricklefs Cancer Research 2016 )
- 6 EVs and the immune response of GBMs: a lot of available research is missing here. Please discuss: van der Vos, Neuro Oncol, 2016, Ricklefs et al. Science Advances 2018, de Vrij Int J Cancer 2015, etc.
- 6. the whole abstract about EVs in GBM metastasis should be deleted or rephrased, because GBM almost never metastasise. They invade. But not metastasise.
- 7. The authors should discuss the main obstacle we face regarding EVs in GBM. The blood brain barrier. There is research tackling this obstacle. Please Revise.
Author Response
Thank you for your review of our paper. We appreciate your help and suggestions. These comments are all valuable and helpful for improving our article. We have modified our manuscript according to your opinions. Please see the attachment.

Reviewer 2 Report
This is an interesting review on the role of exosomes in glioblastoma's pathophysiology with an important clinical impact that needs to draw attention to researchers. I have some minor comments to improve the quality of the article.
- Sections 2 and 3 need to be re-arranged so as to improve the flow of information. First, they should start with discovery of exosomes, then formation and release and then move to structural and functional aspects.
- Sections 6-10 should be subsections of section 5 and be numbered as 5.1-5.5.
- At the conclusion/future perspective section, the authors should also elaborate on targeting options of exosomes and suggest future studies/perspectives.
- Some grammatical mistakes throughout the manuscript need to be corrected.
- Heading of table 1 should be corrected to 'Tumorigenesis in exosomal microRNAs with a role in GBM'
- Heading of table 2 should be corrected to 'Angiogenic microRNAs in exosomes with a role in GBM'
Author Response
Thank you for your professional review work and positive comments on our article. These comments are valuable and helpful for improving our article. We have revised our manuscript according to your opinions, please see the attachment.

Reviewer 3 Report
In this present study, the authors addressed the subject of exosomes in GBM, a very challenging and promising matter, that is progressively growing.
However, the review articles about this subject are flourishing, as DOI: 3390/brainsci10080553 (titled The Involvement of Exosomes in Glioblastoma Development, Diagnosis, Prognosis, and Treatment) or DOI: 10.12659/MSM.924023 (Role of Exosomes in the Progression, Diagnosis, and Treatment of Gliomas) and a very exhaustive review about the crosstalk between glioblastoma cells and astrocytes through the transfer of EV cargo has become an area of interest (DOI: 10.1016/j.tins.2020.10.014)
Then, it is strongly suggested not to replicate what already are available in the literature.
I would rather suggest to the author to reshape the work to make it more original deeply focusing of one aspect of exosome in GBM, as the interaction with microenvironment or the potential use in therapy or how they can represent a source of non-invasive biomarkers
Author Response
Thank you for your professional review and scientific suggestions on our manuscript. Please see the attachment.

Round 2
Reviewer 1 Report
The authors have revised the original manuscript to a sufficient manner, however it needs to be state, that there are already multiple reviews about the role of EVs in GBM and other cancer, so my question really is, how does this review help and add value the scientific world.
Author Response
Many thanks for your positive comments and valuable suggestions to improve the quality of our manuscript. We have modified them seriously and try to explain it according your point. Please see the attachment.

Reviewer 3 Report
The article still remains no original, and it represents one of the multiple review-articles published in these past few years.
Besides, on my previous paper-review I mentioned some articles, that the authors still do not report in the present version.
For the next work, I’ve just point out that the English language is still not fluent and some mistakes are present (as “The TMZ-resistant of GBM cells was related” (Pag 7, row 701), lack of “space” before any bracket).
